# Sex-associated molecular differences for cancer immunotherapy

Youqiong Ye [1,2], Ying Jing[2], Liang Li[3], Gordon B. Mills [4], Lixia Diao [5✉], Hong Liu [6✉] & Leng Han [2,7✉]

Immune checkpoint blockade therapies have extended patient survival across multiple cancer lineages, but there is a heated debate on whether cancer immunotherapy efficacy is different between male and female patients. We summarize the existing meta-analysis to show inconsistent conclusions for whether gender is associated with the immunotherapy response. We analyze molecular profiling from ICB-treated patients to identify molecular differences for immunotherapy responsiveness. We perform comprehensive analyses for patients from The Cancer Genome Atlas (TCGA) and reveal divergent patterns for sex bias in immune features across multiple cancer types. We further validate our observations in multiple independent data sets. Considering that the majority of clinical trials are in melanoma and lung cancer, meta-analyses that pool multiple cancer types have limitations to discern whether cancer immunotherapy efficacy is different between male and female patients. Future studies should include omics profiling to investigate sex-associated molecular differences in immunotherapy.

[1] Shanghai Institute of Immunology, Faculty of Basic Medicine, Shanghai Jiao Tong University School of Medicine, Shanghai 200025, China. [2] Department of Biochemistry and Molecular Biology, The University of Texas Health Science Center at Houston McGovern Medical School, Houston, TX 77030, USA. [3] Department of Biostatistics, The University of Texas MD Anderson Cancer Center, Houston, TX 77030, USA. [4] Knight Cancer Institute, Oregon Health and Science University, Portland, OR 97239, USA. [5] Department of Bioinformatics and Computational Biology, The University of Texas MD Anderson Cancer Center, Houston, TX 77030, USA. [6] Department of Dermatology, Xiangya Hospital, Central South University, Changsha, Hunan 410008, China. [7] Center for Precision Health, The University of Texas Health Science Center at Houston, Houston, TX 77030, USA. ✉email: ldiao@mdaderson.org honyliu1014@csu.edu. cn; leng.han@uth.tmc.edu

I mmune checkpoint blockade (ICB) therapies, including inhibition of programmed cell death 1 (PD-1) or ligand 1 (PD-L1) and cytotoxic T-lymphocyte antigen-4 (CTLA-4), have extended patient survival across multiple cancer lineages[1]. Sex-based immunological differences might have potential impact on immune response[2], whereas its contribution to cancer immunotherapy remains unclear. There is heated debate based on large-scale meta-analysis as to whether cancer immunotherapy efficacy is different between male and female patients[3–6]. Conforti et al.[4,5] reported that male patients treated with immune checkpoint inhibitors achieved greater efficacy than female patients by a meta-analysis of randomized clinical trials. In contrast, two other studies presented conflicting results that there was no significant difference in efficacy between male and female patients treated with immunotherapy[3,6]. These meta-analyses are based on subgroup hazard ratios (HR) of published clinical trials, and may introduce bias due to the lack of analysis of individual patients, and/or features that differ in their distributions between men and women, including smoking behaviors and clinicopathological subtypes[3,7].

Tremendous efforts have been undertaken to identify a series of biomarkers to predict the response to immunotherapy. Tumor with high tumor mutation burden (TMB)[8–11] tend to present more immunogenic neoantigens to enhance the ability of T cells to recognize and kill tumor cells[12]. PD-L1 is actively expressed on both tumor cells and antigen-presenting cells, and inhibition of PD-1 potentially affects multiple steps in the early stage of lymph node and subsequent immune response in the tumor microenvironment[13]. T cell-inflamed gene expression profile (GEP), which includes IFN-γ-related response genes, cytotoxic activity, chemokine expression and adaptive immune resistance, is associated with the response to PD-1 inhibitor pembrolizumab[14]. Cytolytic activity (CYT)[15] can enhance the anti-tumor activity of adaptively transferred T cells, so patients who achieved clinical benefit from ICB therapy had significantly higher CYT than those who had minimal benefit from ICB therapy[10]. Other potential biomarkers have also been reported, including neoantigen load[16,17] and protein expression or mRNA expression of checkpoint mediators (e.g., CTLA-4)[18]. Herein, we performed comprehensive analyses to investigate sex-associated molecular differences of these biomarkers in immune components to better understand gender effects on immunotherapy efficacy.

## Results

**Inconsistent conclusions from meta-analysis.** To address the debate about whether cancer immunotherapy is different between genders, we summarized effect of gender in available data from ICB treatment trials mentioned in previous meta-analyses[3,4] (Supplementary Table 1 and Supplementary Fig. 1). We used the deft approach to calculate trial-specific ratio of HRs and pooled HRs using a random-effects model[19] (see Supplementary methods). We observed an insignificant pooled HR (1.07; 95% confidence interval (CI): 0.95–1.19; random-effects model [REML], $p = 0.28$) by pooling the 27 clinical trials from the two meta-analyses together (Fig. 1). Compared to Conforti et al., Wallis et al. used different selection criteria to add 8 clinical trials (6 out of 8 showed overall survival [OS] advantage in female patients) and remove 4 clinical trials (all with OS advantage in male patients). For each clinical trial assessed separately, we observed that 6 out of 7 clinical trials showed OS advantage in male patients with melanoma. Interestingly, the inconsistent benefit was particularly clear in studies of patients with non-small-cell lung cancer (NSCLC), in that 6 out of 11 clinical trials showed OS advantage in male patients, whereas 5 out of 11 clinical trials showed OS advantage in female patients (Fig. 1). These results

suggested that simply pooling different clinical trials may not provide a definitive result. Furthermore, considering the heterogeneity of control arms of previous meta-analyses, we analyzed the gender effects based on detailed therapies of control arms (Supplementary Table 1), including 5 trials with docetaxel and 4 trials with placebo. We used deft approach to calculate trial-specific ratio of HRs and pooled HRs for immunotherapy vs. docetaxel treatment (HR:1; 95% CI: 0.85–1.2; REML, $p = 0.23$; Supplementary Fig. 2a), and immunotherapy vs. placebo (HR: 0.85; 95% CI: 0.66–1.1; REML, $p = 0.22$; Supplementary Fig. 2b), and again observed the inconsistent benefit for different control arms. In addition, we also did deft analysis for anti-PD-1/PD-L1 and anti-CTLA-4 therapies, respectively. We observed insignificant pooled HR for pooled anti-PD-1/PD-L1 clinical trials (HR: 1.04; 95% CI: 0.90–1.2; REML, $p = 0.61$; Supplementary Fig. 2c) and pooled anti-CTLA-4 clinical trials (HR: 1.20; 95% CI: 0.95–1.51; REML, $p = 0.12$; Supplementary Fig. 2d), potentially due to the inconsistent benefit from each trial. Taken together, our results suggest the difficulty to address this debate by meta-analysis alone.

**Gender-bias of related molecular biomarkers in immunotherapy.** To understand sex-associated molecular mechanisms altering immunotherapy responsiveness, we obtained ICB treatment data sets with molecular profiling for individual patients[9,10,20–24] (Supplementary Table 2). Comparing HR of OS among female patients to male patients with ICB treatment, we observed a divergent pattern (Fig. 2a). Male patients with colorectal cancer (COAD; log-rank test, $p = 0.041$; Supplementary Fig. 3a) or glioblastoma multiforme (GBM; log-rank test, $p = 0.011$; Supplementary Fig. 3b) with anti-PD-1/PD-L1 therapy showed better OS. Female patients with esophagogastric cancer (ESCA) or NSCLC tended to have better OS. We further observed a trend of higher response rate in female patients (16/32 = 50%) compared to male patients (6/24 = 25%) with NSCLC (Supplementary Fig. 3c). The sample size or other confounding factors in these data sets may limit statistical power. We further analyzed molecular biomarkers reported in these immunotherapy data sets, including tumor mutation burden (TMB), individual gene mutation (PBRM1, BRCA2), T cell-inflamed gene expression profile (GEP), neoantigen load, cytolytic activity (CYT) and protein expression of checkpoints (CTLA-4, PD-L1, PD-L2)[9,10,20–24]. We observed significantly higher TMB in male patients with melanoma (Mann–Whitney–Wilcoxon [MWW] test, $p = 0.027$; $p = 0.00039$; Fig. 2b and Supplementary Fig. 3d, e) and bladder cancer (BLCA; MWW test, $p = 0.0024$; Fig. 2b and Supplementary Fig. 3f). Consistent with this, we observed that male patients with BLCA tend to have higher neoantigen load (MWW test, $p = 0.0054$; Fig. 1c and Supplementary Fig. 3g). We observed a significantly higher mutation rate for PBRM1 in male patients with clear cell renal cell carcinoma (ccRCC; MWW test, $p = 0.040$; Fig. 2b; Supplementary Fig. 3h). Other predictors were not significantly different between genders, which might be due to limited sample size.

**A divergent gender-bias of immune features.** We further took advantage of the large sample size and multi-omics data from TCGA[25] and investigated sex-associated molecular differences in immune components for 22 cancer types with ≥20 samples in both female and male groups (Fig. 3a and Supplementary Table 3). To reduce potential confounding effects, we employed a propensity score algorithm, which is an important statistical tool for controlling confounding in observational studies and has been widely used in clinical research[26,27], to reweight potential confounding effect in a multivariate manner[26] (e.g., age at diagnosis,

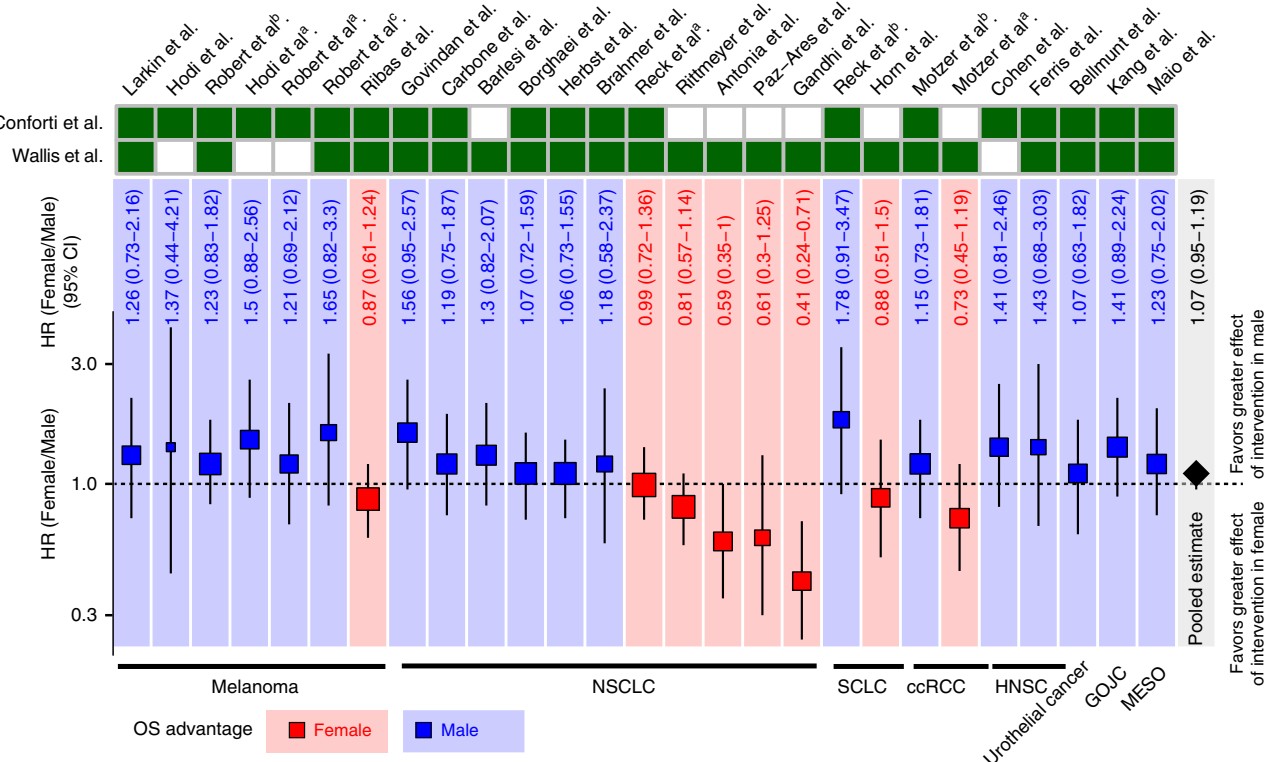

**Fig. 1 Clinical outcomes between male and female patients with ICB treatment.** The interaction between ICB treatment efficacy and gender for clinical trials summarized in Supplementary Fig. 1. Background and square color indicate OS advantage of ICB treatment in female (red) or male (blue). Square size indicates the proportion to the inverse of the variance of the estimates. Black vertical lines indicate the 95% confidence interval (CI). Cells filled with dark green indicates the data set used in this meta-analysis. Sample size for each trial was listed in Supplementary Table 1. [a,b,c]Indicate the different studies from the same first author and published in the same years. ccRCC clear cell renal cell carcinoma, GEJC gastric or gastroesophageal junction carcinoma, HNSC head and neck cancer, NSCLC non-small-cell lung cancer, SCLC, small-cell lung cancer, MESO mesothelioma.

race, smoking status, tumor stage, histological type and tumor purity; see Methods; Supplementary Fig. 4b). We visualized the propensity score distributions in male and female groups, and observed high overlap between male and female in each cancer type (Supplementary Fig. 4c), suggesting the appropriate to using MW method[28]. We included all biomarkers used in patients with ICB treatment (Fig. 3b), as well as other significant biomarkers for immunotherapy[29], including immune cell populations (Fig. 3c), checkpoints (Fig. 3d), TCR/BCR and aneuploidy score (evaluated by somatic copy number variation [SCNV]; Supplementary Fig. 5). Consistent with the ICB treatment data sets, we observed significantly higher TMB (linear regression model, $p = 0.038$; Benjamini and Hochberg correction, FDR = 0.17; Supplementary Fig. 6a; The same statistical analysis for $p$ and FDR was performed in this section), single nucleotide variation neoantigen load ($p = 0.0072$; FDR = 0.12; Supplementary Fig. 6b) and PD-L1 ($p = 0.0036$; FDR = 0.035; Supplementary Fig. 6c) in male patients with melanoma (Fig. 3b). We also observed several other cancer types to have male-biased immune features, including kidney renal papillary cell carcinoma (KIRP), which demonstrated significantly higher TMB ($p = 0.009$; FDR = 0.099; Supplementary Fig. 6d) and CYT ($p = 0.0061$; FDR = 0.099; Supplementary Fig. 6e) (Fig. 3b), higher relative abundance of immune cells (Fig. 3c) and higher mRNA expression of immune checkpoints (Fig. 3d). In contrast, several cancer types showed female-biased immune features. For example, female patients with lung squamous cell carcinoma (LUSC) had significantly higher levels of biomarkers, including CYT ($p = 0.016$; FDR = 0.099; Fig. 3b; Supplementary Fig. 6f), GEP ($p = 0.0017$; FDR = 0.038; Fig. 3b; Supplementary Fig. 6g), relative abundance of

activated CD4[+] T cells and activated CD8[+] T cells (Fig. 3c), 20 out of 34 immune checkpoints (Fig. 3d) and TCR richness ($p = 0.0028$; FDR = 0.062; Supplementary Fig. 6h). We also observed that female patients with LUSC had significantly lower aneuploidy scores than male patients ($p = 1.1 \times 10^{-4}$; FDR = 0.0023; Supplementary Fig. 6i). Interestingly, we observed female-bias for both stimulatory immune checkpoints (e.g., TNFRSF4, TNFSF4, ICOS and CD27) and inhibitory checkpoints (e.g., PDCD1, LAG3, CTLA-4 and ADORA2A). The immune system is complicated and stimulatory immune checkpoints have significant roles in immune activation, while the overexpression of inhibitory immune checkpoints provides the opportunity for other anti-checkpoints and/or combination treatment, thus may improve the clinical outcome of immunotherapy[30]. Nevertheless, our observation suggests a divergent gender-bias of immune features across different cancer types (e.g., male-bias in melanoma vs. female-bias in LUSC).

**Validation of gender-bias in independent data sets.** To validate our observations in TCGA data, we examined multiple additional independent data sets. We observed a male-biased pattern for patients with melanoma based on TMB (Fig. 3b) and confirmed this pattern in two independent melanoma data sets with sample sizes ≥100, with significantly higher TMB (linear regression model, $p = 9.4 \times 10^{-4}$; $p = 0.0031$; Supplementary Fig. 7a). We observed a male-biased pattern for patients with kidney renal clear cell carcinoma (KIRC) in TCGA based on immune checkpoints (e.g., CD28 and CD86) and immune cell populations (e.g., active CD4+ T cells) and confirmed this pattern for patients with the same cancer, labeled ccRCC, in one independent data set[31]

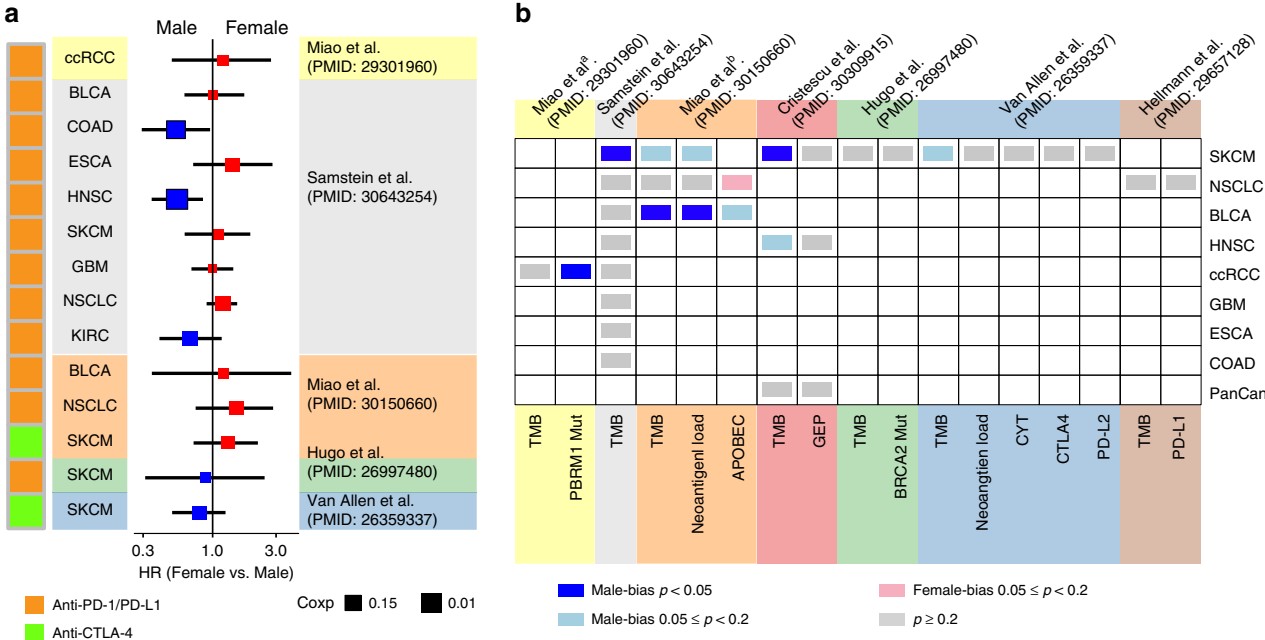

**Fig. 2 Clinical outcomes and molecular differences between male and female patients with ICB treatment. a** Univariate survival analysis in the Cox proportional hazard model for female patients with immunotherapy treatment compared with male patients in nine cancer types from five data sets. Square color indicates OS advantage in female (red) or male (blue). Square size indicates the significance of cox p-value. Black horizontal lines indicate the 95% CI. **b** The gender difference for molecular biomarkers reported for immunotherapy (x axis) across multiple cancer types (y axis) in seven immunotherapy data sets. Two-sided Wilcoxon–Mann–Whitney test was used for continuous variables and Fisher's exact test was used for discrete variables. Blue cell: male-bias with p < 0.05, light blue cell: male-bias with 0.05 ≤ p < 0.2, pink cell: female-bias with 0.05 ≤ p < 0.2. Empty cells indicate the unavailability of data. Sample size for each data set was listed in Supplementary Table 2. ccRCC clear cell renal cell carcinoma, BLCA bladder cancer, COAD colorectal cancer, ESCA esophagogastric cancer, GBM glioma, HNSC head and neck cancer, NSCLC non-small-cell lung cancer, PanCan multiple cancer types with small sample size. PD-L1/PD-L2 programmed cell death ligand 1/2, CTLA-4 cytotoxic T-lymphocyte antigen-4. GEP T cell-inflamed gene expression profile, CYT cytolytic activity.

(Supplementary Fig. 7b). In contrast, we observed a female-biased pattern for patients with lung cancer in TCGA based on immune checkpoints (e.g., BTLA and CD80) and immune cell populations (e.g., activated CD4+/CD8+ T cell) and validated this pattern in one independent data set (GSE47115)[32] (Supplementary Fig. 7b). Our results further confirmed a divergent pattern for sex bias in immune features across multiple cancer types.

## Discussion
Accumulated evidences have demonstrated sex-based differences in immune response involved in autoimmune diseases and response to infections[2], but it is unclear whether these differences contribute to cancer immunotherapy. Recent large-scale meta-analyses have shown contradictory results[3–6]. However, the potential limitations of meta-analysis have been discussed by the research community. For example, meta-analysis relies on published results rather than individual patient data[7], and selection criteria in meta-analysis may lead to contradictory conclusions[33]. Furthermore, it is difficult to control for confounding effects, such as lifestyle or behavior differences between male and female[34]. We summarized the existing meta-analysis to show that IO data may not give a clear conclusion for whether gender is associated with the immunotherapy response. By analyzing molecular profiling in patients with ICB treatment, TCGA patients, as well as several independent data sets, we demonstrated a difference in gender effects between melanoma and NSCLC. These two cancer types were the most common types of cancers included in the meta-analyses, due to the largest number of ICB clinical trials for these cancer types with complete or active status (Fig. 4). Therefore, it may be inappropriate to perform meta-analysis by pooling all cancer types.

Previous studies examined either limited types of immune related biomarkers (e.g., TMB and GEP) or limited number of samples between male and female patients, and reported either male-bias or non-significant pattern[35,36]. In this study, instead of relying on individual biomarkers, we performed a comprehensive analysis of multiple immune features, including immune cell populations, checkpoints, TCR/BCR and aneuploidy score in ICB-treated patients and a large number of TCGA patients, thereby providing a more comprehensive perspective for investigating gender-bias in immunotherapy. We observed an unexpected divergent pattern of sex-associated differences across different cancer types, especially the opposite pattern between patients with melanoma and lung cancer. Our analysis also demonstrated differences in gender for other immune checkpoints (e.g., LAG3 and IDO1) used in clinical trials[37,38], suggesting the future consideration of gender effects in ICB trials. Interestingly, we observed discrepant pattern between the mRNA expression and protein expression of PD-L1, which may due to the moderate correlation of mRNA and protein (Supplementary Fig. 8). Furthermore, we used a rigorous computational approach, the propensity score analysis, to reduce confounding effects for assessment of sex-associated molecular features. Of note, we only considered the confounding factors provided by these data resources, but could not consider other unprovided factors, such as menopausal status. Comprehensive analysis of multi-omics data with rigorous computational approaches will surpass or at least complement the findings from meta-analysis. To the best of our knowledge, our results provide the most comprehensive landscape for addressing the heated debate regarding molecular differences in immunotherapy efficacy between male and female patients with cancer. Our data highlight the importance of

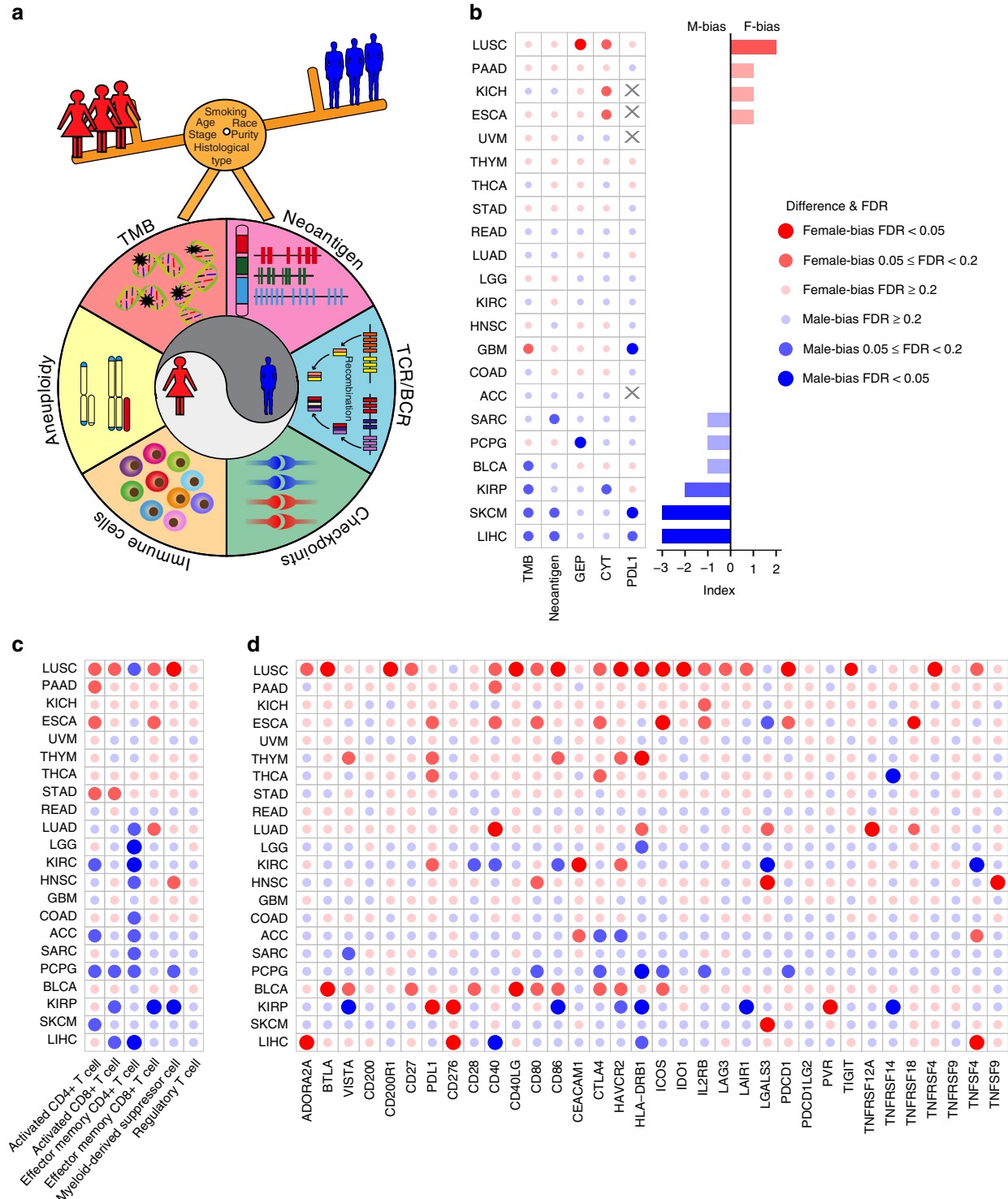

**Fig. 3 Differences in immune features between male and female patients from TCGA. a** Overview of the propensity score algorithm used to balance confounding effects, including age, race, tumor purity, tumor stage, subtype and smoking history, and to evaluate the sex-associated immune features, including TMB, neoantigen load, TCR/BCR, checkpoints, immune cell population and aneuploidy, across cancer types. **b** Differences of molecular biomarkers, including TMB, neoantigen load, GEP, CYT and PD-L1 protein expression, reported in immunotherapy data sets between male patients and female patients. Bar plots indicate the number of significant female-biased features minus the number of significant male-biased features. **c** Differences of relative abundance of six immune cell populations, including active CD4/CD8 T cells, effector memory CD4/CD8 T cells, myeloid-derived suppressor cell and regulatory T cells. **d** Differences of mRNA expression level of 34 immune checkpoints, including LAG3, CTLA-4, PDCD1 and CD274. *X* axis denotes immune features. *Y* axis of **b**–**d** denotes 22 cancer types analyzed by the propensity score algorithm and ordered by the number of significant female-biased features minus the number of significant male-biased features in **b**. Statistical analysis was performed using a propensity score algorithm to identify immune-associated features (see Supplementary methods). *p*-value was calculated by linear regression model and adjusted by Benjamini and Hochberg correction. FDR is labeled as red dots (female-bias) and blue dots (male-bias) in **b**–**d**. Sample size for each data set was listed in Supplementary Table 3.

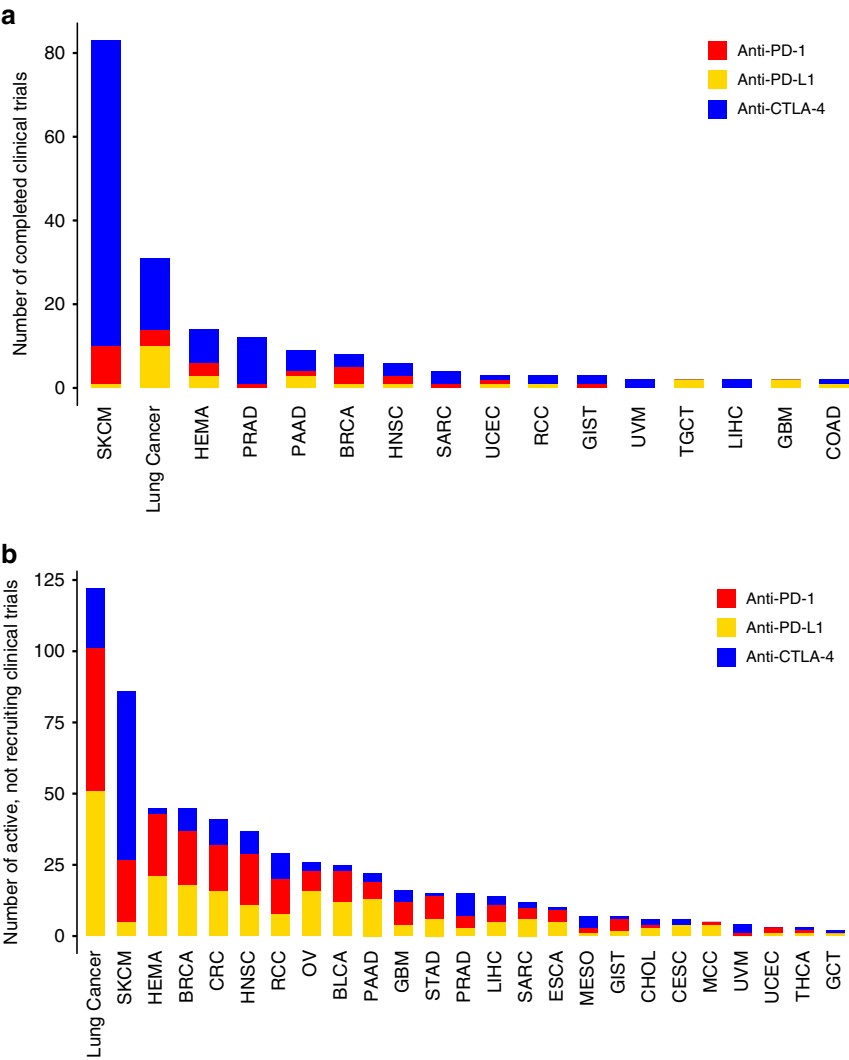

**Fig. 4 Immunotherapy clinical trials across cancer types.** Number of ICB clinical trials for anti-PD-1/PD-L1 and anti-CTLA-4 across cancer types as of April 16, 2019, with status of **a** completed and **b** active, not recruiting.

including molecular profiling data in future clinical trials to understand gender effects on immunotherapy.

## Methods

**Assess the gender effects in clinical trials**. Twenty-seven clinical trials with ICB treatment in eight cancer types (7 trials in melanoma; 11 trials in NSCLC; 2 trials in small-cell lung cancer [SCLC], head and neck cancer [HNSC], and ccRCC; 1 trials in mesothelioma, urothelial cancer and gastric or gastroesophageal junction carcinoma [GOJC];) were obtained from two previous meta-analysis studies[3,4] (detailed information was listed in Supplementary Table 1). We performed deft approach[19]. In brief, we assessed the effect of gender on the immunotherapy efficacy within each trial, and pooled these estimates across trials using random-effects model meta-analysis. These interactions are the difference in the efficacy of ICB treatment between female patients and male patients. Hazard ratio (HR) > 1 indicates OS advantage of ICB treatment in male patients, while HR < 1 indicates OS advantage of ICB treatment in female patients.

**Data analysis of patients with ICB treatment**. We performed comprehensive analyses for immunotherapy data sets with molecular profiling, including four melanoma data sets, two lung cancer data sets, two bladder cancer data sets, two renal cell carcinoma data sets and other cancer types[9,10,20–24] (Supplementary Table 2). Survival analysis was performed by R package survival. HR was calculated by Cox proportional hazards model and 95% CI was reported, and Kaplan–Meier survival curve was modeled by survfit function. The two-sided long-rank test was used to compare Kaplan–Meier survival curves. The comparison of the percentage of benefit and non-benefit between male and female patients was determined by Fisher's exact test. We examined the molecular differences of potential biomarkers reported in these studies for potential mechanisms that alter immunotherapy

responsiveness, including TMB, individual gene mutations (PBRM1, BRCA2), GEP, neoantigen load, CYT and protein expression of checkpoint mediators (CTLA-4, PD-L1, PD-L2). The statistical significance of an individual gene mutation was evaluated by Fisher's exact test and that of other molecular features was assessed using the two-sided Mann–Whitney–Wilcoxon test.

**Data analysis of patients from The Cancer Genome Atlas (TCGA)**. We expanded our analysis to TCGA data[25] for 22 cancer types with ≥20 samples in both female and male groups (Supplementary Table 3). Mutation, gene expression and protein expression were obtained from TCGA (https://portal.gdc.cancer.gov/). The value of aneuploidy, the richness of T cell receptor/B cell receptor (TCR/BCR) and neoantigen load were obtained from Thorsson et al.[39] (https://gdc.cancer.gov/about-data/publications/panimmune). Immune checkpoint genes with known co-stimulatory or co-inhibitory effects in T cells were obtained from Auslander et al.[40] We used gene set variation analysis[41] (GSVA) to compute the relative abundance of the immune cell population and GEP level in each sample based on the gene signatures of six immune cell populations from Charoentong et al.[42] and the GEP gene signature from Ayers et al.[14] CYT was calculated as the geometric mean of the gene expression of two cytolytic markers (GZMA and PRF1)[43]. To balance potential confounding factors, including age at diagnosis, race, smoking status, tumor stage, histological type and tumor purity, between female and male patients, we used the propensity score algorithm[26]. The patient's age and tumor purity are continuous variables, and remained confounding factors are categorical variables. Briefly, we first calculated the propensity score using logistic regression, with sex as the dependent variable, and used matching weight scheme[44] to reweight samples based on their propensity scores. We checked the covariate balance after propensity score weighting using the standardized difference, which is defined by mean covariate difference between the two comparison groups divided by pooled

standard deviation. We then compared the molecular features between these two balanced groups and considered FDR < 0.2 as significance.

**Data analysis of independent data sets**. To further confirm the contextual sex-biased effect, we obtained two independent data sets with TMB for melanoma patients from Australia (Skin Cancer—Australia [MELA-AU]) and Brazil (Skin Adenocarcinoma—Brazil [SKCA-BR]) through the International Cancer Genome Consortium project (https://dcc.icgc.org/). We obtained two independent data sets with gene expression data for patients with lung cancer (GSE47115)[32] and clear cell renal cell carcinoma (ccRCC) patients (GSE73731)[31]. Statistical analysis was performed using propensity score algorithm described above, and we considered FDR < 0.2 as significance.

**Number of ICB clinical trials**. We searched "immune checkpoint blockade clinical trials" across all cancer types on ClinicalTrials.gov by April 16, 2019 for status as completed and as active, not recruiting. We retained cancer types with at least two clinical trials. We classified treatment strategies, including anti-PD-1 (pembrolizumab, nivolumab, cemiplimab), anti-PD-L1 (atezolizumab, avelumab, durvalumab) and anti-CTLA-4 (ipilimumab, tremelimumab), in each cancer type.

**Reporting summary**. Further information on research design is available in the Nature Research Reporting Summary linked to this article.

## Data availability

All associated data are available in Source Data 1 and 2 for Figs. 1–4 and Supplementary Figs. 1–8, respectively. All the remaining data are available within the Article, Supplementary Information files or available from the author upon reasonable request.

## Code availability

Codes were implemented in R 3.6.2 and are deposited in https://github.com/youqiongye/SexImm.

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

## Acknowledgements

This work was supported by the Cancer Prevention and Research Institute of Texas (Grant Nos. RR150085 and RP190570) to CPRIT Scholar in Cancer Research (L.H.). This work was supported by the National Cancer Institute (Grant No. P30CA016672) to L.L. We thank LeeAnn Chastain for editorial assistance.

## Author contributions

L.H. conceived and supervised the project. Y.Y. and L.H. designed and performed the research. Y.J., L.D. and L.L. contributed to the data analysis. Y.Y., G.B.M., L.D., H.L. and L.H. interpreted the results and wrote the manuscript.

## Competing interests

G.B.M. has sponsored research support from AstraZeneca, Critical Outcomes Technology, Karus, Illumina, Immunomet, Nanostring, Tarveda and Immunomet and is on the Scientific Advisory Board for AstraZeneca, Critical Outcomes Technology, ImmunoMet, Ionis, Nuevolution, Symphogen and Tarveda. All other authors declare no competing interests.
