## [Peer Review File · Nature Communications]

Reviewers' comments:

Reviewer #1 (Remarks to the Author):

The revised report is improved in terms of clarity of purpose and contribution of the authors. The previous presentation of the meta analysis and associated interpretation was clearly confusing and now improved.

Is there a way to differentiate whether the various sex-based differences that are found within these datasets are attributable to sex as opposed to idiosyncratic M:F differences that may be incidentally found when examining so many comparisons?

Is it statistically appropriate to include the p/FDR 0.05-0.20 group as a "trend" bias as opposed to NS?

Could effect-size rather or some absolute determination of group-based differences rather than p-values to used to visualize meaningful differences in groups?

Reviewer #2 (Remarks to the Author):

I thank the authors for their answers. They have addressed my concerns. Having read, as well, their answers to the other reviewers-- and while not speaking for those reviewers -- I feel the authors addressed those concerns.

My feeling is this: These are difficult analyses to do and if we hold up publication in good journals of this kind of comprehensive work then the field overall will suffer.

For me, the paper is good for acceptance.

#1 comments to authors:

The revised report is improved in terms of clarity of purpose and contribution of the authors. The previous presentation of the meta-analysis and associated interpretation was clearly confusing and now improved.

Response: We thank the reviewer for positive comments.

Is there a way to differentiate whether the various sex-based differences that are found within these datasets are attributable to sex as opposed to idiosyncratic M:F differences that may be incidentally found when examining so many comparisons?

Response: We thank the reviewer for this valuable comment. For analysis with large number of comparisons (e.g., Fig. 3), we performed Benjamini & Hochberg correction to carefully control the false discovery rate for multiple comparisons, which are normally utilized for such situations¹⁻¹⁴. For the datasets with few comparisons, we did not apply Benjamini & Hochberg correction as other studies¹⁵⁻¹⁸, and we collected multiple datasets for analysis (e.g., Fig. 2b). Furthermore, the consistent results among different datasets suggesting that our observation is unlikely to be introduced by many comparisons.

Is it statistically appropriate to include the p/FDR 0.05-0.20 group as a "trend" bias as opposed to NS?

Response: P-value is the probability of obtaining test results at least as extreme as the results actually observed during the test, assuming that the null hypothesis is correct. FDR is used to assess the false discovery rate, and many studies considered Benjamin-Hochberg (BH) FDR < 0.2 as statistical significance¹⁹⁻²⁸. We believe our analysis is statistically appropriate.

Could effect-size rather or some absolute determination of group-based differences rather than p-values to used to visualize meaningful differences in groups?

Response: Thank you so much for reviewer's great suggestion. For different types of molecular features, the group-based differences have wide range, makes it challenge to display differences in one panel. For example, the mean weighted log2-transformation of TMB ranges from 3.50 to 8.21, and the log2-fold change for TMB ranges from -0.77 to 0.34; the mean weighted log2-transformation neoantigen load ranges from 2.26 to 7.9, and the log2-fold change for neoantigen load ranges from -2.07 to 2.86; The mean of weighted GEP range from -0.17 to 0.12, and the differences range from -0.19 to 0.16. In general, the comparisons with larger difference will have more significant FDR. Therefore, we only displayed FDR in the figure. Per the reviewer's suggestion, we listed absolute determination of group-based differences in the newly added **Source Data 1**.

#2 comments to authors:

I thank the authors for their answers. They have addressed my concerns. Having read, as well, their answers to the other reviewers-- and while not speaking for those reviewers -- I feel the authors addressed those concerns. My feeling is this: These are difficult analyses to do and if we hold up publication in good journals of this kind of comprehensive work then the field overall will suffer. For me, the paper is good for acceptance.

Response: We thank the reviewer for positive comments.

References

1. Benjamini, Y. & Hochberg, Y. Controlling the False Discovery Rate: A Practical and Powerful Approach to Multiple Testing. *Journal of the Royal Statistical Society: Series B (Methodological)* (1995). doi:10.1111/j.2517-6161.1995.tb02031.x
2. Mascaux, C. *et al.* Immune evasion before tumour invasion in early lung squamous carcinogenesis. *Nature* **571**, 570–575 (2019).
3. Berger, A. C. *et al.* A Comprehensive Pan-Cancer Molecular Study of Gynecologic and Breast Cancers. *Cancer Cell* 690–705 (2018). doi:10.1016/j.ccell.2018.03.014
4. Bailey, M. H. *et al.* Comprehensive Characterization of Cancer Driver Genes and Mutations. *Cell* **173**, 371–385.e18 (2018).
5. Wang, Z. *et al.* lncRNA Epigenetic Landscape Analysis Identifies EPIC1 as an Oncogenic lncRNA that Interacts with MYC and Promotes Cell-Cycle Progression in Cancer. *Cancer Cell* 706–720 (2018). doi:10.1016/j.ccell.2018.03.006
6. Karamitros, D. *et al.* Single-cell analysis reveals the continuum of human lympho-myeloid progenitor cells article. *Nature Immunology* **19**, 85–97 (2018).
7. Zhao, J. *et al.* Immune and genomic correlates of response to anti-PD-1 immunotherapy in glioblastoma. *Nature Medicine* **25**, 462–469 (2019).
8. Hu, X. *et al.* Landscape of B cell immunity and related immune evasion in human cancers. *Nature Genetics* **51**, 560–567 (2019).
9. Parikh, K. *et al.* Colonic epithelial cell diversity in health and inflammatory bowel disease. *Nature* **567**, 49–55 (2019).
10. Ordovas-Montanes, J. *et al.* Allergic inflammatory memory in human respiratory epithelial progenitor cells. *Nature* **560**, 649–654 (2018).
11. Crinier, A. *et al.* High-Dimensional Single-Cell Analysis Identifies Organ-Specific Signatures and Conserved NK Cell Subsets in Humans and Mice. *Immunity* **49**, 971–986.e5 (2018).
12. Ruan, H. *et al.* Comprehensive characterization of circular RNAs in ~1000 human cancer cell lines. *Genome Medicine* **11**, 1–14 (2019).
13. Campbell, J. D. *et al.* Genomic, Pathway Network, and Immunologic Features Distinguishing Squamous Carcinomas Graphical. *Cell Reports* 194–212 (2018). doi:10.1016/j.celrep.2018.03.063
14. Zhou, Z. Y. *et al.* Genome wide analyses uncover allele-specific RNA editing in human and mouse. *Nucleic acids research* **46**, 8888–8897 (2018).
15. Hu, Q. *et al.* Oncogenic lncRNA downregulates cancer cell antigen presentation and intrinsic tumor suppression. *Nature Immunology* **20**, 835–851 (2019).
16. Fairfax, B. P. *et al.* Peripheral CD8+ T cell characteristics associated with durable responses to immune checkpoint blockade in patients with metastatic melanoma. *Nature*

- Medicine* **26**, 193–199 (2020).
17. Mahmoudi, E., Kiltchewskij, D., Fitzsimmons, C. & Cairns, M. J. Depolarization-Associated CircRNA Regulate Neural Gene Expression and in Some Cases May Function as Templates for Translation. *Cells* **9**, 25 (2019).
 18. Ya, R. M. *et al.* ImmuCellAI: a unique method for comprehensive T-cell subsets abundance prediction and its application in cancer immunotherapy. *Advance Science* 201902880 (2019). doi:10.1101/872184
 19. Dorval, V. *et al.* Gene and microRNA transcriptome analysis of Parkinson's related LRRK2 mouse models. *PLoS ONE* **9**, e85510 (2014).
 20. Sanli, K., Karlsson, F. H., Nookaew, I. & Nielsen, J. FANTOM: Functional and taxonomic analysis of metagenomes. *BMC Bioinformatics* **14**, 38 (2013).
 21. Seumois, G. *et al.* Transcriptional Profiling of Th2 Cells Identifies Pathogenic Features Associated with Asthma. *The Journal of Immunology* **197**, 655–664 (2016).
 22. Klarić, L. *et al.* Glycosylation of immunoglobulin G is regulated by a large network of genes pleiotropic with inflammatory diseases. *Science Advances* **6**, eaax0301 (2020).
 23. Wu, J. *et al.* Magnetic resonance imaging and molecular features associated with tumor-infiltrating lymphocytes in breast cancer. *Breast Cancer Research* **20**, 101 (2018).
 24. Wang, J. *et al.* LAT, HOXD3 and NFE2L3 identified as novel DNA methylation-driven genes and prognostic markers in human clear cell renal cell carcinoma by integrative bioinformatics approaches. *Journal of Cancer* **10**, 6726–673 (2019).
 25. Miao, X., Luo, Q., Qin, X., Guo, Y. & Zhao, H. Genome-wide mRNA-seq profiling reveals predominant down-regulation of lipid metabolic processes in adipose tissues of Small Tail Han than Dorset sheep. *Biochemical and Biophysical Research Communications* **467**, 413–420 (2015).
 26. Schüssler-Fiorenza Rose, S. M. *et al.* A longitudinal big data approach for precision health. *Nature Medicine* **25**, 792–804 (2019).
 27. Diss, G. & Lehner, B. The genetic landscape of a physical interaction. *eLife* **7**, e32472 (2018).
 28. Han, S. *et al.* Genome-wide association study of childhood acute lymphoblastic leukemia in Korea. *Leukemia Research* **34**, 1271–1274 (2010).